# A Non-Nutritive Feeding Intervention Alters the Expression of Efflux Transporters in the Gastrointestinal Tract

**DOI:** 10.3390/pharmaceutics13111789

**Published:** 2021-10-26

**Authors:** Yang Mai, Francesca K. H. Gavins, Liu Dou, Jing Liu, Farhan Taherali, Manal E. Alkahtani, Sudaxshina Murdan, Abdul W. Basit, Mine Orlu

**Affiliations:** 1School of Pharmaceutical Sciences (Shenzhen), Sun Yat-sen University, Guangzhou 510275, China; maiy6@mail.sysu.edu.cn (Y.M.); liu.dou.14@alumni.ucl.ac.uk (L.D.); liuj663@mail2.sysu.edu.cn (J.L.); 2UCL School of Pharmacy, University College London, 29-39 Brunswick Square, London WC1N 1AX, UK; francesca.gavins.13@ucl.ac.uk (F.K.H.G.); farhan.taherali.15@alumni.ucl.ac.uk (F.T.); manal.alkahtani.17@ucl.ac.uk (M.E.A.); s.murdan@ucl.ac.uk (S.M.); a.basit@ucl.ac.uk (A.W.B.)

**Keywords:** food effect, sex differences, hormones, efflux pumps, preclinical drug development, oral delivery, food–drug interactions

## Abstract

Intestinal interactions with nutrients, xenobiotics and endogenous hormones can influence the expression of clinically relevant membrane transporters. These changes in the gastrointestinal (GI) physiology can in turn affect the absorption of numerous drug substrates. Several studies have examined the effect of food on intestinal transporters in male and female humans and animal models. However, to our knowledge no studies have investigated the influence of a non-nutritive fibre meal on intestinal efflux transporters and key sex and GI hormones. Here, we show that a fibre meal increased the acute expression of P-glycoprotein (P-gp), breast cancer resistance protein (BCRP), and multidrug-resistance-associated protein-2 (MRP2) in small intestinal segments in both male and female Wistar rats. Enzyme-linked immunosorbent assays were used for the protein quantification of efflux transporters and hormonal plasma concentration. In male rats, the fibre meal caused the plasma concentration of the GI hormone cholecystokinin (CCK) to increase by 75% and the sex hormone testosterone to decrease by 50%, whereas, in contrast, the housing food meal caused a decrease in CCK by 32% and testosterone saw an increase of 31%. No significant changes in the hormonal concentrations, however, were seen in female rats. A deeper understanding of the modulation of efflux transporters by sex, food intake and time can improve our understanding of inter- and intra-variability in the pharmacokinetics of drug substrates.

## 1. Introduction

Oral dosage forms are the most preferred route of administration due to patient preference and pharmaceutical manufacturing capabilities. Drug absorption through the intestinal tract is a highly complex process, influenced by numerous internal physiological factors, ranging from gastrointestinal (GI) motility, pH profile, the presence of enzymes and bile salts and the mucus layer [1,2]. In addition, patient-specific factors such as sex, age, disease-state and diet can considerably affect drug pharmacokinetics [3]. Therefore, predicting biopharmaceutics processes can be challenging [4]. Importantly, over the last 10 years, the labelling of 40% of approved oral drugs have reported that food significantly influences their absorption [5]. As the effects of food on drugs can occur at different stages of drug processing in the body [6], the key mechanism for such effects is often unknown [7,8,9]. Furthermore, sex differences exist in the body’s handling of food, which in turn can affect drug performance between males and females in the postprandial state [10]. For example, sex differences in the bioavailability of cyclosporine A, a P-glycoprotein (P-gp) substrate, were reported after a fat-rich meal. In females, a decreased oral bioavailability was found, whereas an increased oral bioavailability was shown in male humans [11].

Rats are commonly used animal models in oral drug development to predict human pharmacokinetic responses to drug products [12]. Importantly, there are similarities between the rat’s and the human’s GI physiology; both show small intestinal transit times of 3–4 h and rates of gastric emptying that are controlled by the energy content of the ingested food [13,14,15]. Our group has previously shown similar sex differences in the GI P-gp expression of fasted male and female rats and humans [16], suggesting that the rat is an appropriate animal model to predict the drug absorption of P-gp substrates in humans.

At the intestinal layer, efflux transporters shuttle substrates from enterocytes into the GI lumen, providing a protective barrier and limiting their absorption into the systemic circulation. Substrates can include several endogenous compounds (sterols, bile acids and hormones), nutrients (sugars, fatty acids and vitamins) and drug products [17]. In particular, P-gp (MDR1, *ABCB1*), breast cancer resistance protein (BCRP, *ABCG2*), and multidrug-resistance-associated protein 2 (MRP2, *ABCC2*) can contribute to poor absorption and low bioavailability of many drugs [18]. Dou et al., observed that food consumption resulted in a change in the intestinal relative and absolute P-gp expression to different extents in male and female rats [19,20]. However, the mechanism for the food-mediated phenomenon was poorly understood. We hypothesized that the sex-dependent food effect in the modulation of P-gp may be multifactorial; an interplay between food components in a meal and the release of GI hormones and sex hormones [19]. Firstly, several components in food are shown to modulate the intestinal absorption of P-gp substrates [21]. Dietary fibre can beneficially modulate GI activity by altering transit time and stool formation in humans [22]. In addition, dietary fibres can also alter the expression and functions of intestinal P-gp and BCRP in humans [23], cell-lines and male rats [24]. Secondly, GI hormones may increase P-gp membrane localization [25]. Thirdly, testosterone, the primary male sex hormone, and estradiol, a female sex hormone, were shown to inhibit or induce P-gp, respectively [26,27]. Our previous studies and the literature predominantly focuses on the expression of P-gp, whereas other intestinal transporters such as BCRP and MRP2 play significant roles in drug absorption.

To address these questions of the sex-dependent food effects, our study aimed to investigate three feeding interventions—fasted, housing food (normal meal) and a non-nutritive substance (fibre meal) in male and female Wistar rats. The expression of three efflux transporters, P-gp, BCRP and MRP2 were evaluated in the small intestine (duodenum, jejunum and ileum) as well as the plasma concentration over time (0 to 2 h) for the GI hormone cholecystokinin (CCK) and the sex hormones testosterone and estradiol.

## 2. Materials and Methods

### 2.1. Materials

Krebs-bicarbonate Ringer’s solution (KBR), pH 7.4, was freshly prepared before the experiment at room temperature and was kept at 37 °C. KBR was composed of 10 mM D-glucose, 1.2 mM CaCl_2_, 1.2 mM MgCl_2_, 115 mM NaCl, 25 mM NaHCO_3_, 0.4 mM KH_2_PO_4_ and 2.4 mM K_2_HPO_4_ [28]. Lysis buffer was freshly prepared with 50 mM Tris, 250 mM NaCl, 5 mM ethylenediaminetetraacetic acid (EDTA), 1 mM Na_3_VO_4_, 1 mM phenylmethylsulfonyl fluoride (PMSF), 1% Nonidet P40 and protease inhibitor cocktail from Sigma (Darmstadt, Germany) in a phosphate-buffered saline (PBS) solution and stored at 4 °C.

ELISA kits were purchased from MEIMAN Biotech (Guangzhou, China). The ELISA kits were as follows: rat P-gp ELISA Kit (MM-0604R2), rat BCRP ELISA kit (MM-0606R2), rat MRP2 ELISA kit (MM-0607R2), rat CCK ELISA Kit (MM-0034R2), rat testosterone ELISA kit (MM-0577R1) and rat estradiol ELISA kit (MM-0567R2). Cellulose pellets (Solka-Floc^®^ 200 FCC) were purchased from Envigo–Teklad custom diet (TD.85467) (Madison, WI, USA). Normal meal was housing food provided in the animal housing unit.

### 2.2. Animals

Male and female Wistar rats (healthy, 8 to 13 weeks old) were used as the animal models and each experimental unit used 6 rats. The rats were housed at room temperature (25 °C) in a light–dark cycle of 12 h. The rats acclimatized to the animal unit for at least 7 days. An overnight fast of 12 h was conducted prior to the experiments the following morning at 8 am. All animal work was conducted in accordance with the project license (8002536), approved by the Home Office under the Animals (Scientific Procedures) Act 1986 on 7 June 2012. The fasted group was the control group and the animals were sacrificed at t = 0 h. The rats were administered the normal meal and fibre meal suspensions by oral gavage and were sacrificed at t = 0.5 h, t = 1 h and t = 2 h. A timeframe of 2 h was chosen for animal welfare reasons after the overnight fast. The suspensions were prepared in the concentration of 0.125 g/mL by mixing 0.5 g of meal (normal meal and fibre meal) with 4 mL deionized water. Fibre meal was composed of crushed cellulose pellets (Section 2.1), which were non-fermentable in the GI tract. A description of the fibre meal with images is provided in the Appendix A. The normal meal was composed of corn, wheat middling, wheat, soybean meal, Peru fish meal, chicken meal, premixed materials, limestone, wheat gluten and soybean oil.

The rats were sacrificed by CO_2_ asphyxiation. Blood was taken by cardiac puncture and kept on ice until further preparation. Their intestines were immediately excised and stored in an ice-cold KBR solution. Roughly 1 cm pieces of the small intestine; duodenum (1 cm from the ligament of Treitz), jejunum (10 cm from the ligament of Treitz) and ileum (1 cm from the cecum) were opened along their mesenteric border. The tissues were gently washed with KBR solution to remove the intestinal contents.

### 2.3. Transporter Extraction from the Jejunum

The mucosal tissues (approximately 50 mg) were cut into small pieces and homogenized in 0.5 mL RIPA lysis buffer at 30 Hz for 30 s with a TissueLyser (QIAGEN, Hilden, Germany), and repeated twice at intervals of 30 s for complete homogenisation. The tissue homogenates were incubated at 4 °C for 2 h, and then centrifuged at 12,000× *g* for 5 min. The total tissue protein was collected in the supernatants, and its concentration was subsequently determined with the Pierce™ BCA protein assay kit (Beyotime Biotechnology, Shanghai, China) according to the manufacturer’s instructions. To measure the targeted transporter protein level, P-gp, BCRP and MRP2 were quantified by ELISA kits (Meimian Biotech, Guangzhou, China) based on the manufacturer’s description.

### 2.4. Preparation of Hormone Blood Samples

Blood samples were centrifuged at 10,000 rpm for 10 min within 24 h of sampling. The supernatant (plasma samples) was collected and placed in a 1.5 mL Eppendorf tube and immediately stored at −20 °C prior to analysis. To measure the hormone levels, CCK, testosterone and estradiol concentrations were quantified by ELISA kits based on the manufacturer’s description.

### 2.5. Data Analysis and Visualisation

The results generated in the study were expressed as mean ± standard deviation (SD) (*n* = 6). The data were analysed by a one-way analysis of variance (ANOVA) in each segment, followed by a Tukey post-hoc analysis with a 95% confidence interval using GraphPad Prism (version 9, GraphPad Software, San Diego, CA USA). Pearson correlations (r) were also calculated in GraphPad Prism. All plots were constructed in Python version 3.9.0 (Dover, DE, USA) on Jupyter Notebook version 6.0.3 (San Diego, CA, USA) using the Matplotlib package version 3.4.3 [29].

## 3. Results

### 3.1. Efflux Transporter Expression

#### 3.1.1. P-Glycoprotein (P-gp)

The changes in P-gp expression with the feeding interventions are illustrated in Figure 1a,b. In the fasted state, a significant sex difference was reported; the P-gp expression was 17% higher in male rats than in females (Appendix A). On the other hand, in the normal meal fed state, a contrasting trend was seen where the P-gp expression was 102% higher in female rats than male rats. Interestingly, in the fibre meal group, male rats showed a significantly higher P-gp expression than in the normal meal group, 18.54 ± 0.38 and 8.58 ± 0.23 ng/mg, respectively, whereas for female rats, the P-gp expression was not significantly different between the normal meal and fibre meal groups, with values of 17.30 ± 1.32 and 16.68 ± 1.33 ng/mg, respectively.

Figure 1c reports the influence of food interventions on P-gp expression over time. Significant sex differences were seen in the jejunum with all feeding interventions (fasted, normal meal and fibre meal). An interesting trend was observed in male rats, where the normal meal feeding intervention caused a decrease in P-gp expression from 12.4 ± 0.70 to 8.6 ± 0.23 ng/mg. In contrast, the fibre meal caused an increase in P-gp expression from 12.4 ± 0.7 to 18.5 ± 0.38 ng/mg. On the other hand, in females, P-gp expression increased for both food interventions in the first hour. Between 1 to 2 h, the P-gp expression decreased in female rats with the normal meal from 17.3 ± 1.32 to 12.97 ± 1.17 ng/mg, whereas in the fibre meal intervention, the P-gp expression increased from 16.68 ± 1.33 to 18.38 ± 0.90 ng/mg.

#### 3.1.2. Breast Cancer Resistant Protein (BCRP)

Figure 2a, b reports a statistically significant difference in BCRP expression between the fasted and the fibre meal interventions in both sexes along the small intestine (Appendix A). Sex differences were seen in the duodenum for all feeding interventions. Female rats showed a higher level of expression in the duodenum; 9.32 ng/mg, 11.40 ng/mg and 12.08 ng/mg, compared to 8.24 ng/mg, 8.79 ng/mg and 11.15 ng/mg, for the fasted, normal meal and fibre meal groups, respectively. Furthermore, Figure 2c shows that the fibre meal intervention significantly increased the jejunal BCRP levels after 30 min, 50% in males and 57% in females. In contrast, for the normal meal group, the female BCRP levels increased by 19% in the first 30 min and then gradually decreased after two hours. The male normal meal intervention showed the greatest increase (18%) between the 1 and 2 h time interval.

#### 3.1.3. Multidrug-Resistance-Associated Protein 2 (MRP2)

MRP2 expression in male and female rats along the small intestine is reported in Figure 3a,b and Appendix A. Significant differences (*p* < 0.05) were reported between the fasted and fibre meal interventions in both sexes along the small intestine; the largest increase was 103% in the male jejunum. Interestingly, the normal meal caused an increase in the MRP2 expression of male rats: 1.6-fold in the duodenum, 1.7-fold in the jejunum and 1.5-fold in the ileum, compared to the fasted state. However, this housing food-mediated effect was not seen in female rats. Figure 3c shows a similar jejunal MRP2 expression at 30 min in the fibre meal and male normal meal interventions. Significantly, following normal meal intake, the female MRP2 expression was lower than in males. In addition, the female MRP2 expression in the normal meal intervention was similar (4.23 ± 0.57 ng/mg at 0 h, 4.46 ± 0.49 ng/mg at 0.5 h, 4.28 ± 0.72 ng/mg at 1 h and 4.51 ± 0.57 ng/mg).

### 3.2. Hormone Concentration

#### 3.2.1. Gastrointestinal Hormone: Cholecystokinin

Figure 4 displays the cholecystokinin (CCK) plasma concentration (pg/mL) over time in hours in male and female rats under fasted state and fed (normal meal and fibre meal) states. A significant sex difference was seen in the fasted state at t = 0, in the normal meal state at each time points and in the fibre meal fed state at 2 h. Interestingly, a clear statistical difference (*p* < 0.05) was seen between the normal meal and the fibre meal intervention in male rats. An increase of 75% was seen in the CCK plasma concentration between 0 h and 0.5 h in the fibre meal intervention, with a decrease of 32% in the normal meal intervention.

#### 3.2.2. Sex Hormone Concentrations: Testosterone and Oestradiol

Interestingly, the basal testosterone was shown to be higher in female rats than in male rats. Additionally, in female rats, the testosterone concentration decreased until 1 h for the normal meal intervention, then increased (Figure 5a), whereas the testosterone continued to decrease with the fibre meal. A contrasting trend was observed for testosterone concentration between the feeding groups in male rats; the normal meal showed an increase of 23% up to 2 h and the fibre meal showed a decrease of 54%. The basal estradiol was higher in females than males. Significant sex differences (*p* < 0.05) were seen for the estradiol concentration (Figure 5b), but the type of meal did not affect the plasma concentration.

### 3.3. Correlation between Efflux Transporter Expression and Hormone Expression

Figure 6 explores the correlation between the expression of P-gp, BCRP and MRP2 transporters and the concentration of testosterone, estradiol and CCK hormones. The highest correlations were the negative correlations shown between testosterone and P-gp (r = −0.99, r = −0.99, r = −0.99 and r = −0.92 for males and females in the normal and fibre meals, respectively). For BCRP, testosterone showed a strong negative correlation with the female fibre group (r = −0.99) and estradiol showed a strong positive correlation with the male normal group (r = 0.89). CCK concentration was moderately correlated with male P-gp expression (r = 0.81 and 0.75 for normal and fibre, respectively). In addition, estradiol was moderately correlated with MRP2 in the female normal meal group (r = 0.85).

## 4. Discussion

Both food, sex, time and hormone concentration acutely modulated the expression of intestinal efflux transporters in male and female rats. Importantly for drug delivery, intestinal transporters are key determinants of the pharmacokinetics of many drugs [30]. In fact, P-gp, BCRP and MRP2 are considered to be clinically relevant for the oral absorption of drugs by the International Transporter Consortium [31]. BCRP is a half-size transporter [32], in contrast to P-gp and MRP2. Drug substrates of P-gp include digoxin, fexofenadine and paclitaxel; for BCRP, they include rosuvastatin, sulfasalazine and doxorubicin; and for MRP2 they include indinavir and cisplatin. However, inter- and intra-variability exists in their expression and activity with implications for drug substrate absorption. A key external factor influencing the variabilities is food consumption [3]. In food effect studies, specific high-fat diets have been designed to maximize the potential for drugs to show a food effect [33] and are used in human, dog and pig models [34,35]. Whereas in studies using small rodents, housing food is often used to study the fed state [19]. To the authors’ knowledge, the effects of a non-nutritive type of fibre on efflux transporters have not been thoroughly explored by in vivo male and female animal model studies.

An interesting phenomenon was observed with the fibre meal intervention in all of the efflux transporters. Its presence increased transporter expression in both sexes across the small intestine. The greatest percentage increase was seen in the P-gp expression in the female jejunum (+58%) for the fibre meal. In the male jejunum and ileum, increases of 50% and 40% from the fasted state were reported, respectively. A suggestion for these results could be that the presence of insoluble foodstuff, the fibre cellulose, will add bulk to the lumen of GI tract and contribute to the increased gastric luminal pressure, distension of the stomach, delay in gastric emptying and release of GI hormones. The body may protect itself by increasing the expression of the efflux transporters as a barrier mechanism to prevent the absorption of potentially toxic ingested compounds. The normal meal, on the other hand, will be digested in the GI tract and broken down into products of digestion and then key nutrients will be absorbed.

Key sex differences were reported in the expression of P-gp and MRP2 in the jejunum with the normal meal intervention. In fed male rats, a lower expression of P-gp (−45%) was found in the jejunum, compared with the fasted rats. In females, on the other hand, a higher expression of P-gp (+64%) was shown, compared with the fasted state. For MRP2, a sex difference was reported in the jejunum for the normal meal and in the duodenum with the fasted state. The normal meal caused an increase from the fasted state for males (+50%), but not for females, where jejunal MRP2 expression was 4.23 ± 0.57 and 4.28 ± 0.72 ng/mg for the fasted and normal meal interventions, respectively. The fasted and housing food (normal meal) P-gp-related findings were similar to previous findings in our research group by Dou et al. using a validated LC–MS/MS method [19], a Western blot method [20] and PCR [16]. Importantly, the inter-individual variations were low in this study, shown by narrow standard deviation in the Appendix A, in comparison to the Western blotting and LC–MS/MS methods [16,19,20].

However, differing results are found in the literature. Dahan and Amidon, using a Western blot method, found an increase in the P-gp expression level in the distal ileum compared with the proximal jejunum in the fasted male Wistar rats [36]. MacLean et al. reported no sex differences in P-gp, BCRP and MRP2 in fed male and female Han-Wistar rats using immunohistochemistry [37]. In addition, MacLean’s study showed the relative P-gp expression increased from proximal to distal regions, BCRP showed an arcuate pattern with highest expression toward the end of the small intestine, and MRP2 decreased along the intestinal axis from proximal to distal parts [37]. Our study showed that for BCRP and MRP2, the expression exhibited a statistically significant increase between the duodenum and jejunum for the fibre meals and the male fasted and normal meal groups. No statistically significant increases were seen for the fasted or normal meal female rats. The aforementioned studies used Western blotting methods which depend on an internal standard protein and the relative abundance may vary between the intestinal segments. However, in the present work an ELISA method was used with specific antibodies. Our findings are similar to our group’s recent article [16]. In the male fasted state, there was a statistically significant difference between the duodenum and jejunum and no statistically significant difference between the jejunum and the ileum. In contrast, in the female fasted state, there was no statistically significant difference between the duodenum, jejunum and ileum.

Fast and reversible modulation of the activity of efflux transporters is of interest to formulation scientists and pharmacologists for the drug delivery of transporter substrates [38]. Changes to the transporter expression can be regulated by (i) changes in their protein expression (long-term regulation) or (ii) changes that do not modify the total amount of protein (acute regulation) [38], by transcriptional or post-transcriptional changes. Interestingly, clear changes are seen in the P-gp, BCRP and MRP2 expression 30 min after the feeding interventions. Potent inhibitors for BCRP, MRP2 and P-gp have been described in the literature [39]. These findings suggest that a dosage form could be designed which releases a specific transporter inhibitor to inhibit their expression [40], followed by a drug substrate. Short-term reversibility of the transporter could ensure the normal physiological functioning of the intestinal tract after the dose has been absorbed [38]. A proposed mechanism for P-gp inhibition is the inhibition of ATPase activity, as P-gp is an energy-activated protein. The substrate binds to the binding site, and then ATP binds to the two binding sites in the nucleotide-binding domains for ATP. This is followed by hydrolysis of ATP, which causes conformational change, where the substrate is released from the protein. The second molecule of ATP is hydrolysed, allowing a conformational reset and the substrate and ATP can bind again [41]. In addition, MRP2 and BCRP are shown to be ATP-dependent transporters for the cellular extrusion of their substrates [42,43]. In clinical development, an application for oral paclitaxel (P-gp substrate) called Oraxol is being filed with the United States Food and Drug Administration (FDA) by the company Athenex. Here, the dosing requires that patients are in the fasted state and take one tablet of the P-gp inhibitor encequidar. Then, one hour later, patients take multiple capsules of paclitaxel as a bio-enhanced formulation [44]. This case study illustrates the clinical importance of the acute response of efflux transporters.

A study from Yano and colleagues found that the CCK hormone levels increased the P-gp localisation and transporter activity in Caco-2 cells [25]. Furthermore, a study from Karhunen et al. found that fibre caused an increase in CCK in humans [45]. CCK has been shown to delay gastric emptying to reduce the amount of nutrients entering the intestines. Prolonging the gastric residence may be one of the stomach’s responses to a non-nutritive substance so that the maximum amount of nutrients can be extracted by the digestive enzymes. The inhibitory effect of CCK on food intake, which acts by reducing meal size and duration, is short-lived, lasting less than 30 min [46]. In this study, a sharp rise in CCK over 30 min, is reported in male rats in the fibre group followed by a subsequent decrease, reflecting the short-lived nature of this hormone. Furthermore, CCK can increase satiation to terminate feeding and reduce meal duration as a result of the presence of “foodstuff” in the GI tract [45].

In the normal meal intervention in males, a decrease in the P-gp expression was observed (Figure 1a). A contributing factor to this change in P-gp could be that a component in the food could modulate the expression of P-gp in male but not in female rats. The rodent diet is rich in phytoestrogens, including genistein, a component in soy (soybean is listed in the normal meal, Section 2.2). Arias et al., reported that ethynylestradiol and genistein, associated with soy ingestion, influenced the expression and activity of MRP2, P-gp and BCRP transporters in the Caco-2 cell model (a male cell line) [47]. In addition, the testosterone concentration decreased in the male normal meal (Figure 5a) and testosterone was highly correlated to P-gp expression (Figure 6). Testosterone administration was reported to decrease the functionality of P-gp in male rabbits [48]. Therefore, the drop in male testosterone may be related to the decrease in P-gp. However, interestingly, the testosterone concentration increased in the male fibre meal group.

The normal meal-mediated increase in P-gp in females could be explained by the female innate protection in the intestinal epithelial layer. No significant hormonal changes were seen in the female rats in the two feeding interventions. Higher levels of CCK were found in the female rats, with no significant difference between the normal meal and fibre meal. Bitter ligands, which could have been present in the case of the cellulose fibre meal, can increase the release of the satiety hormone CCK [49]. As shown in Figure 5a, the basal testosterone in females were higher than in males. This could be a contributing factor for females showing a lower P-gp expression in the fasted state compared to males.

The influence of a fibre meal showed sex differences in the testosterone and CCK hormone concentrations. Additional studies are required to explore the change in transporter functionality between the sexes by assessing substrate activity using different concentrations of a P-gp inhibitor. Furthermore, the acute effects of different diets could be explored by investigating the influence of the meal components (fats, proteins and carbohydrates) on the GI physiology to understand how these food types will affect the delivery of drug products.

## 5. Conclusions

This study has reported the P-gp, BCRP and MRP2 expression along the intestinal tract under three acute feeding interventions: fasting, fed with a normal meal and fed with fibre meal, in male and female Wistar rats. The fibre meal increased the P-gp, BCRP and MRP2 transporter expression along the small intestinal tract in both sexes. Changes in testosterone and CCK concentrations were observed in the male rat with the fibre meal compared to the normal housing meal, but not in the female rat. Key sex differences were seen in the jejunal P-gp and MRP2 expression with normal meal. Therefore, our findings suggest that the products of digestion may modulate P-gp and MRP2 expression in a sex-dependent manner. This is the first study demonstrating that a non-nutritive fibre meal influences transporter expression in the GI tract of male and female rats.

## Figures and Tables

**Figure 1 pharmaceutics-13-01789-f001:**
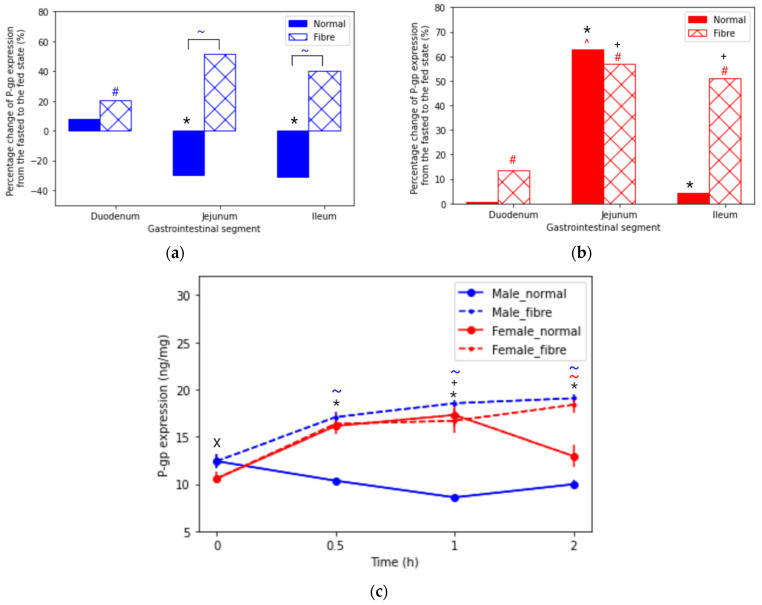
Percentage change in P-glycoprotein (P-gp) expression across the intestinal tract in (**a**) male and (**b**) female rats at time = 1 h from the fasted state to the fed states (normal meal and fibre meal). (**c**) P-gp expression (ng/mg) in the jejunum from time 0 to 2 h under fasted and fed states (normal meal and fibre meal) (mean ± SD) (*n* = 6). The following symbols denote a statistical significance (*p* < 0.05) showing a sex difference between male and female rats: fasted state (X), normal meal (*) and fibre meal (+); and a food effect between the feeding interventions: fasted and fibre (#), fasted and normal (^) and normal and fibre (~).

**Figure 2 pharmaceutics-13-01789-f002:**
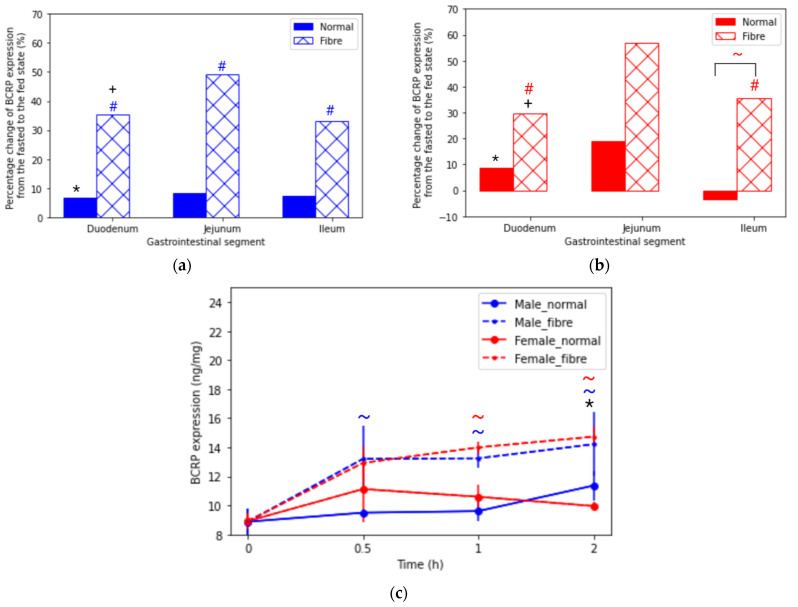
Percentage change in breast cancer resistant protein (BCRP) expression across the intestinal tract in (**a**) male and (**b**) female rats at time = 1 h from the fasted state to the fed states (normal meal and fibre meal). (**c**) BCRP expression in the jejunum from time 0 to 2 h under fasted and fed states (normal meal and fibre meal) (mean ± SD) (*n* = 6). The following symbols denote a statistical significance (*p* < 0.05) showing a sex difference between male and female rats: normal meal (*) and fibre meal (+); and a food effect between the feeding interventions: fasted and fibre (#) and normal and fibre (~).

**Figure 3 pharmaceutics-13-01789-f003:**
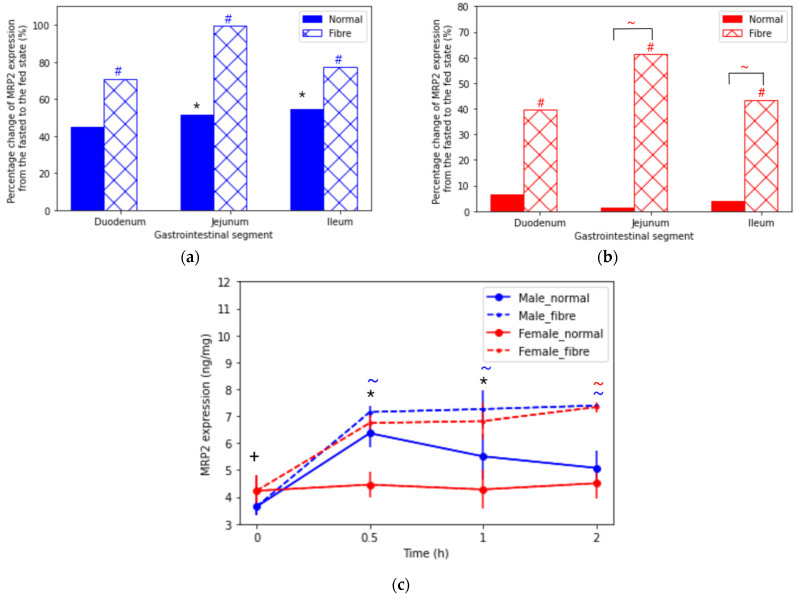
Percentage change in multidrug-resistance-associated protein 2 (MRP2) expression across the intestinal (GI) tract in (**a**) male and (**b**) female rats at time = 1 h from the fasted state to the fed states (normal meal and fibre meal) and (**c**) in the jejunum from time 0 to 2 h under fasted and fed states (normal meal and fibre meal) (mean ± SD) (*n* = 6). The following symbols denote a statistical significance (*p* < 0.05) showing a sex difference between male and female rats: normal meal (*) and fibre meal (+); and a food effect between the feeding interventions: fasted and fibre (#)and normal and fibre (~).

**Figure 4 pharmaceutics-13-01789-f004:**
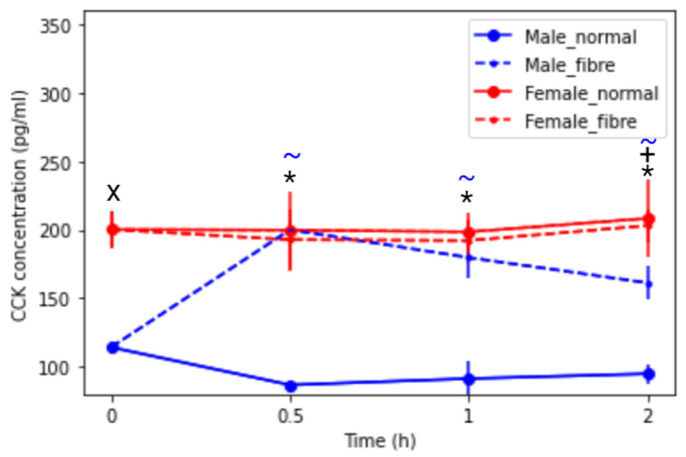
Cholecystokinin concentration (pg/mL) over time of male and female rats under fasted and fed states (normal housing food and fibre meal), (mean ± SD, *n* = 6). The following symbols denote a statistical significance (*p* < 0.05) showing a sex difference between male and female rats: fasted state (X), normal meal (*) and fibre meal (+); and a food effect between the feeding intervention: normal and fibre (~).

**Figure 5 pharmaceutics-13-01789-f005:**
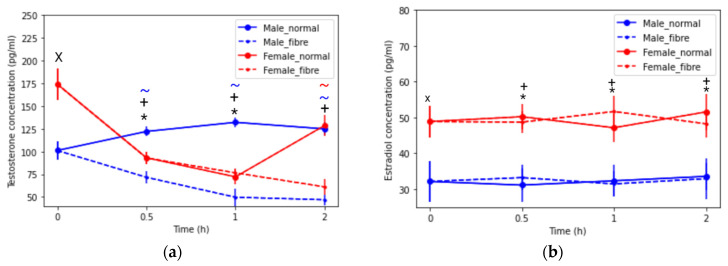
Plasma (**a**) testosterone concentration (pg/mL) and (**b**) estradiol concentration (pg/mL) over time (0 to 2 h) in male and female rats under fasted and fed states (normal meal and fibre meal) (mean ± SD, *n* = 6). The following symbols denote a statistical significance (*p* < 0.05) of a sex difference between male and female rats: fasted state (X), normal meal (*) and fibre meal (+); and a food effect between the feeding interventions: normal and fibre (~).

**Figure 6 pharmaceutics-13-01789-f006:**
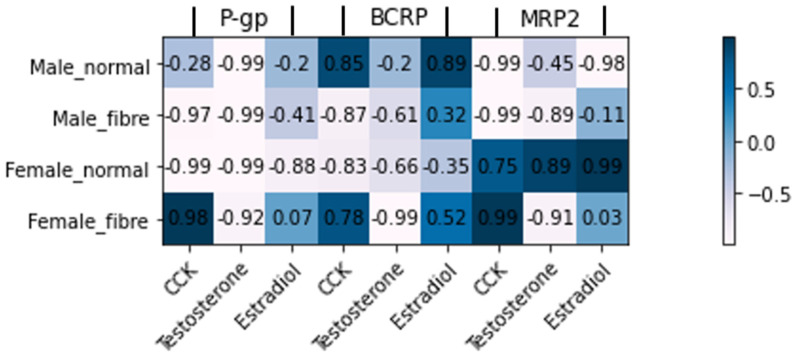
Correlation between transporter and hormone concentration for: P-glycoprotein (P-gp), breast cancer resistant protein (BCRP), multidrug-resistance-associated protein 2 (MRP2) expression in the jejunum with the cholecystokinin (CCK) gastrointestinal hormone and testosterone and estradiol sex hormones.

## Data Availability

The data presented in this study are available in this article and the Appendix A.

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
