# Peer review of "A Non-Nutritive Feeding Intervention Alters the Expression of Efflux Transporters in the Gastrointestinal Tract"

_pharmaceutics, 2021, doi:10.3390/pharmaceutics13111789_

Round 1

Reviewer 1 Report

This study is very interesting, and investigating intestinal interactions and consequent transporter modulation by food/xenobiotics,  is very important in the light of drug absorption. Nevertheless, I do have some concerns and questions regarding the study setup, design and discussion. 

How do you explain the timeframe between the 'fake food' administration and rat sacrifice, from 1/2h to 2h. How reliable is this timeframe in the light of your conslusions and how does it compare to previous studies? How fast can the P-gP expression change?

I believe it is more relevant to follow chronic exposure to these foods on the transporter expression than acute. 

Why do you think these results are different from previous reports which state that the P-gP expression increases from proximal to distal SI (including your own publication http://dx.doi.org/10.1021/acs.molpharmaceut.0c00574)?

How do you explain the choice of fake food? 

Were the investigated samples solely mucosal tissues, or entire intestinal segment including muscle layer? If it were only mucosal tissues, describe the process of mucosal extraction.

Improve formatting of references. 

The graphical abstract is missing despite being mentioned in the paper. 

Reviewer 2 Report

The authors describe the effects of what they call "fake food" intake on the expression of transporters in the GI tract, and compare it to the effect of normal food intake.  The "fake food" is comprised of cellulose pellets, so in reality it is more akin to a "fiber meal" or "cellulose meal".  This is can potentially be a very high impact study, and the authors make an important set of observations, because fiber intake is a medically important variable in the human diet, Indeed, fiber intake can affect many aspects related to gastrointestinal motility (in humans, fiber is used as a laxative -"Metamucil" and a stool softener).  I suggest the authors look into whether anything is known about fiber intake and gastrointestinal transporter expression and perhaps drug absorption in both rats and humans, and then include some background information related to that in the introduction.  

In terms of improving the manuscript, I would suggest that the authors do not refer to "fake food" and instead refer to it as "cellulose meal" or "fiber meal" (I suggest that, to keep things consistent, that the authors use the term  "normal meal" instead of normal food). 

My other suggestion is that the authors take a low magnification microscope picture of the fiber meal that they are using, and then describe it in terms of the size, shape and general features of the particles that make up the meal. Do the particles swell when exposed to water? I suggest looking at the swelling properties of the particles.  I also suggest looking at the stability properties of the particles when they are stirred in a shaker, to see if the particles are stable or if they disintegrate. Certainly, it is possible that such physical features of the particles may influence the phenomena they are studying. 

In order to determine whether the effects they are reporting are due to volume of the fiber meal or perhaps due to some other property (e.g. intestinal distension due to water intake or abrasion of the intestinal wall), I would suggest the authors do a control experiment with inert glass beads of comparable dimensions to their the particles in the fiber meal, to see if the glass beads have the same or different effect. 

Round 2

Reviewer 1 Report

Thank you for replying to my comments. 

Please elucidate in detail the reason for investigation and the clinical importance of acute response of efflux transporters, since they are going back to baseline rather fast (also, do you know how fast?). 
